# Transcranial magnetic stimulation (TMS) inhibits cortical dendrites

Sean C Murphy[1,2,3]*, Lucy M Palmer[3], Thomas Nyffeler[4,5,6], René M Müri[5,6], Matthew E Larkum[1,2]*

[1]Neurocure Cluster of Excellence, Humboldt University, Berlin, Germany; [2]Physiologisches Institut, Universität Bern, Bern, Switzerland; [3]Florey Institute of Neuroscience and Mental Health, University of Melbourne, Melbourne, Australia; [4]Neurology and Neurorehabilitation Centre, Luzerner Kantonsspital, Luzern, Switzerland; [5]Departments of Neurology, University Hospital, Inselspital, University of Bern, Bern, Switzerland; [6]Department of Clinical Research, University Hospital, Inselspital, University of Bern, Bern, Switzerland

**Abstract** One of the leading approaches to non-invasively treat a variety of brain disorders is transcranial magnetic stimulation (TMS). However, despite its clinical prevalence, very little is known about the action of TMS at the cellular level let alone what effect it might have at the subcellular level (e.g. dendrites). Here, we examine the effect of single-pulse TMS on dendritic activity in layer 5 pyramidal neurons of the somatosensory cortex using an optical fiber imaging approach. We find that TMS causes $GABA_B$-mediated inhibition of sensory-evoked dendritic $Ca^{2+}$ activity. We conclude that TMS directly activates fibers within the upper cortical layers that leads to the activation of dendrite-targeting inhibitory neurons which in turn suppress dendritic $Ca^{2+}$ activity. This result implies a specificity of TMS at the dendritic level that could in principle be exploited for investigating these structures non-invasively.

*For correspondence: sean.murphy@florey.edu.au (SCM); matthew.larkum@gmail.com (MEL)

**Competing interests:** The authors declare that no competing interests exist.

## Introduction

Transcranial magnetic stimulation (TMS) holds great promise as a non-invasive method that can be used to both enhance and impair cognitive abilities (*Eldaief et al., 2013*). As such, it has proved to be an important tool for addressing basic questions about brain function as well as for diagnostic and therapeutic purposes (*Fregni and Pascual-Leone, 2007*). Stimulation is produced by generating a brief, high-intensity magnetic field by passing a brief electric current through a magnetic coil. As a general rule, TMS affects the action of feedback projections (*Juan and Walsh, 2003*; *Hung et al., 2005*; *Camprodon et al., 2010*; *Zanto et al., 2011*) leading to a disruption in perception (*Shimojo et al., 2001*; *Kammer et al., 2005*). Due to this influence on higher order cognitive processing, TMS is not only useful for examining the interactions of different brain areas (*Pascual-Leone and Walsh, 2001*; *Silvanto et al., 2005*; *Murd et al., 2012*), but it has also been used as a therapeutic method to alleviate some of the symptoms of hemispatial neglect (*Nyffeler et al., 2009*), schizophrenia including auditory hallucinations (*Giesel et al., 2012*), pain, depression and epilepsy. Despite great interest (*Mueller et al., 2014*), the cellular influence of TMS has yet to be ascertained since the precise effects of TMS at the level of a single neuron are very difficult to gauge, particularly in humans.

The neural architecture of the brain means the neural processes which receive and transform most synaptic inputs, the dendrites, extend into the upper layers where TMS would be most effective. Since dendrites can shape synaptic input to be greater or less than their linear sum (*Polsky et al., 2004*; *Losonczy and Magee, 2006*; *Larkum et al., 2009*) thereby altering the firing

**eLife digest** The brain's billions of neurons communicate with one another using electrical signals. Applying a magnetic field to a small area of the scalp can temporarily disrupt these signals by inducing small electrical currents in the brain tissue underneath. The currents interfere with the brain's own electrical signals and temporarily disrupt the activity of the stimulated brain region.

This technique, which is known as transcranial magnetic stimulation, is often used to investigate the roles of specific brain regions. By examining what happens when a region is briefly taken 'offline', it is possible to deduce what that area normally does. Transcranial magnetic stimulation is also used to treat brain disorders ranging from epilepsy to schizophrenia without the need for surgery or drugs. But despite its widespread usage, little is known about how transcranial magnetic stimulation affects individual neurons.

Neurons are made up of a cell body, which has numerous short branches called dendrites, and a cable-like structure called the axon. Neurons signal to each other by releasing chemical messengers across junctions called synapses. The chemical signals are generally released from the axon of one neuron and bind to receptor proteins on a dendrite on another neuron to stimulate electrical activity in the receiving neuron.

Murphy et al. have now investigated how transcranial magnetic stimulation affects the activity of dendrites from neurons within the cortex of the rat brain. This revealed that the magnetic fields stimulate other neurons that inhibit the activity of dendrites from neurons within the deeper layers of the cortex. The inhibition process depends on a type of receptor protein in the dendrites called $GABA_B$ receptors; blocking these receptors prevents transcranial magnetic stimulation from altering the activity of stimulated brain regions.

The processes occurring in these dendrites have been linked to cognitive function. The next challenge will be to integrate the non-invasive transcranial magnetic stimulation approach with cognitive tests in humans that can now manipulate dendritic activity to test their importance under various circumstances.

properties of the neuron (*Larkum et al., 1999*), the active properties of dendrites have attracted attention and have been linked to cognitive processes and feature selectivity (*Lavzin et al., 2012*; *Xu et al., 2012*; *Smith et al., 2013*; *Cichon and Gan, 2015*). Furthermore, it has been suggested that active dendritic processing underlies a more general principle of cortical operation that facilitates parallel associations between different cortical regions and the thalamus (*Larkum, 2013*) which is controlled by dendritic inhibition (*Palmer et al., 2012*; *Lovett-Barron and Losonczy, 2014*). Establishing the validity of this hypothesis will have important ramifications for understanding brain function as a whole. TMS presents a most promising way to study the causal relationship between active dendritic properties and cognition but only if its effect on dendrites can be understood and ultimately controlled.

Using an optical fiber imaging approach, here we present a study examining the effect of TMS on sensory-evoked dendritic activity in layer 5 pyramidal neurons of the somatosensory cortex. We find that TMS suppresses dendritic $Ca^{2+}$ activity evoked by tactile stimulation and that this suppression can be abolished by blocking $GABA_B$ receptors and excitatory transmission in the upper layers of the cortex. We uncover the cellular mechanisms underlying TMS-evoked inhibition, demonstrating that TMS of the rat brain activates long-range fibers that leads to the activation of dendrite-targeting inhibitory neurons in the upper cortical layers which in-turn suppress dendritic $Ca^{2+}$ activity. Since indirect brain stimulation shows immense promise in treating many neurological disorders, such as epilepsy (*Berenyi et al., 2012*), this study not only illustrates the cellular mechanisms underlying TMS but also highlights dendrites as potential targets for therapeutic approaches.

## Results

We recorded the $Ca^{2+}$ activity in populations of layer 5 (L5) pyramidal neuron dendrites in the hindlimb somatosensory cortex of urethane anesthetized rats using a custom-made fiber optic 'periscope' in vivo (*Murayama et al., 2007*) oriented horizontally for use in tandem with a TMS coil

(*Figure 1A*, *Figure 1—figure supplement 1A*). Pyramidal neurons located approximately 800 µm below the cortical surface were loaded with the $Ca^{2+}$ indicator Oregon Green BAPTA1 AM (OGB1 AM; *Figure 1A* inset and see 'Materials and methods'). Using this approach, large dendritic $Ca^{2+}$ responses to brief hindpaw stimulation (100 V, 1 ms) were reliably evoked after 70 min loading with OGB1 AM (*Figure 1—figure supplement 1B*). To investigate the effects of TMS on evoked cortical network activity, the TMS coil was positioned just above the craniotomy (*Figure 1A*) and a single brief TMS pulse was evoked together with the stimulation of the hindpaw (*Figure 1B*) greater than 70 min post loading with OGB1 AM. TMS caused a significant decrease in the hindpaw-evoked dendritic $Ca^{2+}$ response when triggered 50 ms before hindpaw stimulation (*Figure 1C* and *Figure 1—figure supplement 2A*), both in the maximum amplitude (control, 7.3 ± 1.5 △F/F versus TMS, 5.0 ± 1.1 △F/F, n = 17, p<0.05) and integral (control, 4.3 ± 0.9 △F/F•s versus TMS, 2.4 ± 0.6 △F/F•s; n = 17; p<0.001, *Figure 1D*). Further, the size of the coil (*Figure 1—figure supplement 2B*) and the type of hindpaw stimulation (*Figure 1—figure supplement 2C*) did not influence the results, whereas the distance of the coil from the cortical region of interest influenced the effectiveness of the TMS inhibition on the dendritic sensory responses (*Figure 1—figure supplement 3*).

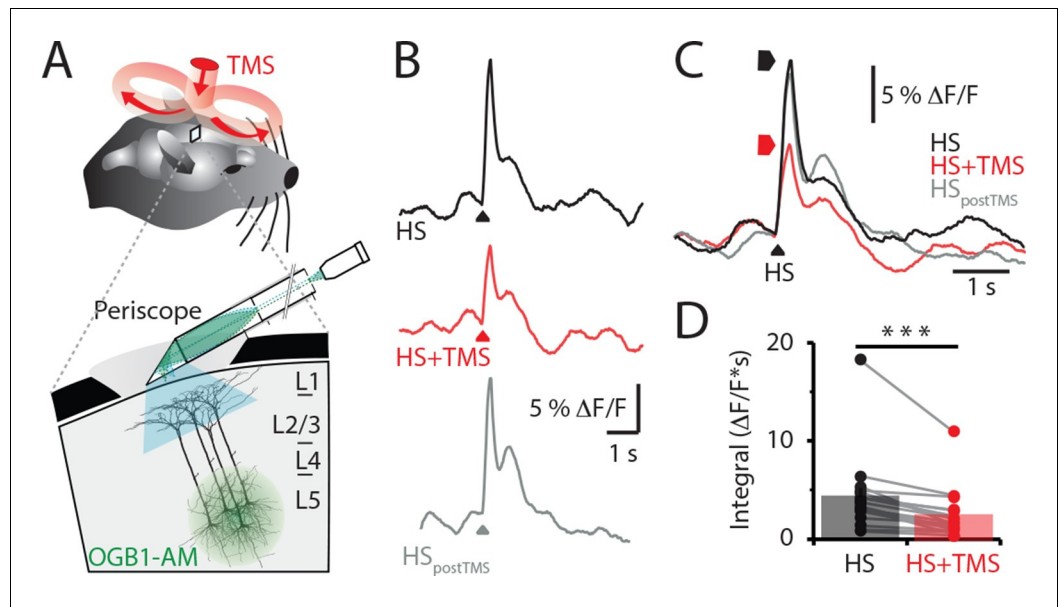

**Figure 1.** TMS inhibits sensory evoked $Ca2^+$ activity in layer 5 dendrites. (**A**) Schematic of the experimental design. Layer 5 pyramidal neurons were bulk loaded with OGB1-AM and dendritic $Ca^{2+}$ activity was recorded using a flat-periscope configured horizontally and inserted underneath the TMS coil from the side. The TMS coil was placed above the dendrites in the hindpaw region of the somatosensory cortex. (**B**) Typical dendritic $Ca^{2+}$ response to hindpaw stimulation (HP) alone (black) and during a single TMS pulse (red) and HP alone post-experiment (grey). (**C**) Overlay of traces in (b) and (**D**) graph illustrating the decrease in $Ca^{2+}$ response during TMS (n=17). p<0.001 (***). TMS, transcranial magnetic stimulation.

The following source data and figure supplements are available for figure 1:

**Source data 1.** Integral and amplitude of evoked calcium transient.

**Figure supplement 1.** Periscope position and temporal characteristics of sensory-evoked $Ca2^+$ responses in layer 5 pyramidal neuron dendrites.

**Figure supplement 2.** The effect of TMS timing, coil size and stimulation paradigm on layer 5 dendritic sensory responses.

**Figure supplement 3.** Increasing TMS strength did not elicite an excitatory response in layer 5 pyramidal neuron dendrites.

What is the cause of this TMS-induced decrease in dendritic calcium activity? L5 pyramidal neuron dendrites have been previously shown to be strongly inhibited by the activation of post-synaptic GABA$_B$ (*Pérez-Garci et al., 2006*; *Chalifoux and Carter, 2011*; *Palmer et al., 2012*) receptors. To test whether GABA$_B$ receptors are predominantly causing the TMS-induced dendritic inhibition, the GABA$_B$ antagonist CGP52432 was locally perfused into the recording region (*Figure 2A*) affecting up to 300 µm of the surrounding tissue (*Figure 2—figure supplement 1*). Blocking GABA$_B$ receptors prevented the TMS-evoked decrease in the Ca$^{2+}$ response to hindpaw stimulation in both the integral (control$_{cgp}$, 2.3 ± 0.5 △F/F•s versus TMS$_{cgp}$, 2.1 ± 0.5 △F/F•s; p=0.62) and maximum amplitude (control$_{cgp}$, 7.7 ± 2.8 △F/F versus TMS$_{cgp}$, 7.2 ± 1.9 △F/F; n = 7; p=0.66, *Figure 2B*). L5 dendrites have been shown to also be inhibited by the activation of GABA$_A$ (*Kim et al., 1995*; *Murayama et al., 2009*) receptors. Although cortical application of Gabazine causes a six-fold increase in the sensory evoked dendritic response (*Figure 2—figure supplement 2*), block of GABA$_A$ receptors also prevented the TMS-evoked decrease in Ca$^{2+}$ response to hindpaw stimulation (HS amplitude, 130 ± 30% of control; n = 6; *Figure 2—figure supplement 2*). Taken together, the fact that blocking both GABA$_A$ and GABA$_B$ receptors abolished the dendritic effect of TMS s.

To investigate the laminar profile of the influence of TMS on neuronal activity, the Ca$^{2+}$ indicator OGB1-AM was injected at different cortical depths (L5, 800 µm; Layer 2/3, 300 µm; Layer 1, 100 µm) and the Ca$^{2+}$ response to TMS alone was recorded. TMS itself did not directly activate L5 pyramidal neuron dendrites (*Figure 3A*; n = 4), contrasting greatly to the large TMS-evoked Ca$^{2+}$ response in cells within layer 2/3 (L2/3; 3.0 ± 1.1 △F/F; n = 3; *Figure 3B*) and layer 1 (L1; 5.5 ± 2.1 △F/F; n = 12; *Figure 3B*). Importantly, the TMS-evoked Ca$^{2+}$ response in these upper cortical layers was similar to the response evoked by physiological stimulation via hindpaw stimulation (L2/3, 3.5 ± 1.2 △F/F•s and L1, 6.4 ± 2 △F/F•s). The lack of a direct response to TMS in L5 pyramidal neuron dendrites implies that the inhibition of sensory evoked dendritic transients was mediated by the action of inhibitory neurons. Furthermore, the response to TMS in upper-layer neurons leaves open the possibility that local inhibitory neurons might be recruited by TMS either directly, via TMS-induced membrane activation or indirectly, via synaptic transmission.

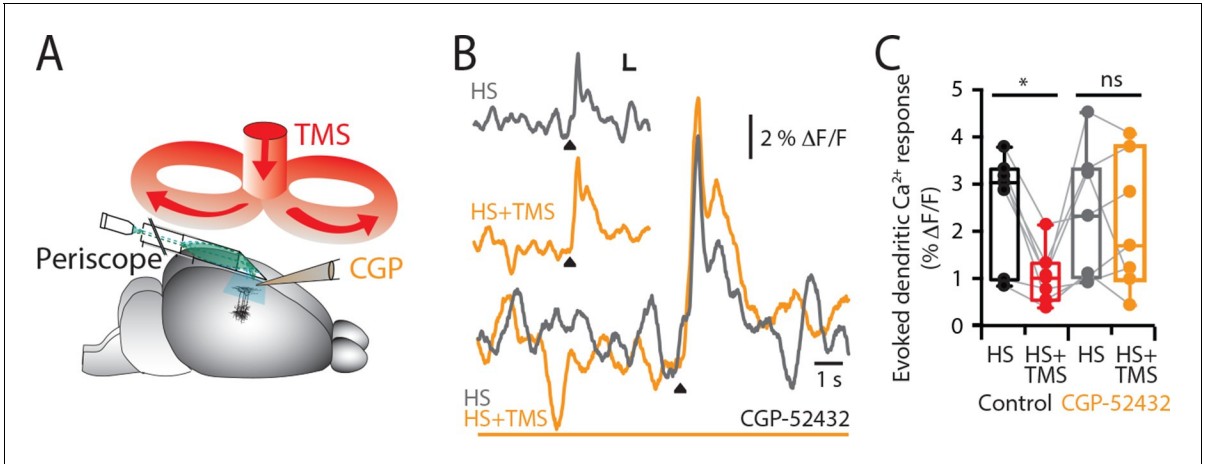

**Figure 2.** TMS causes GABA$_B$-mediated inhibition of layer 5 dendrites. (**A**) Schematic of the experimental design illustrating the application of the GABA$_B$ antagonist CGP52432 on the cortical surface. (**B**) Typical dendritic Ca$^{2+}$ response to hindpaw stimulation (HS) alone (grey) and during a single TMS pulse (orange, HS+TMS) during cortical CGP. (**C**) Block of TMS-evoked inhibition of the dendritic sensory response in the presence of CGP52432 compared with control (prior to CGP52432; HS, black; HS+TMS, red; n=7). p<0.05 (*).

The following figure supplements are available for figure 2:

**Figure supplement 1.** Spread of localized injection (**A–C**) and cortical surface application (**D–F**).

**Figure supplement 2.** Cortical application of the GABAA antagonist Gabazine causes a dramatic increase in the dendritic response to hindpaw stimulation.

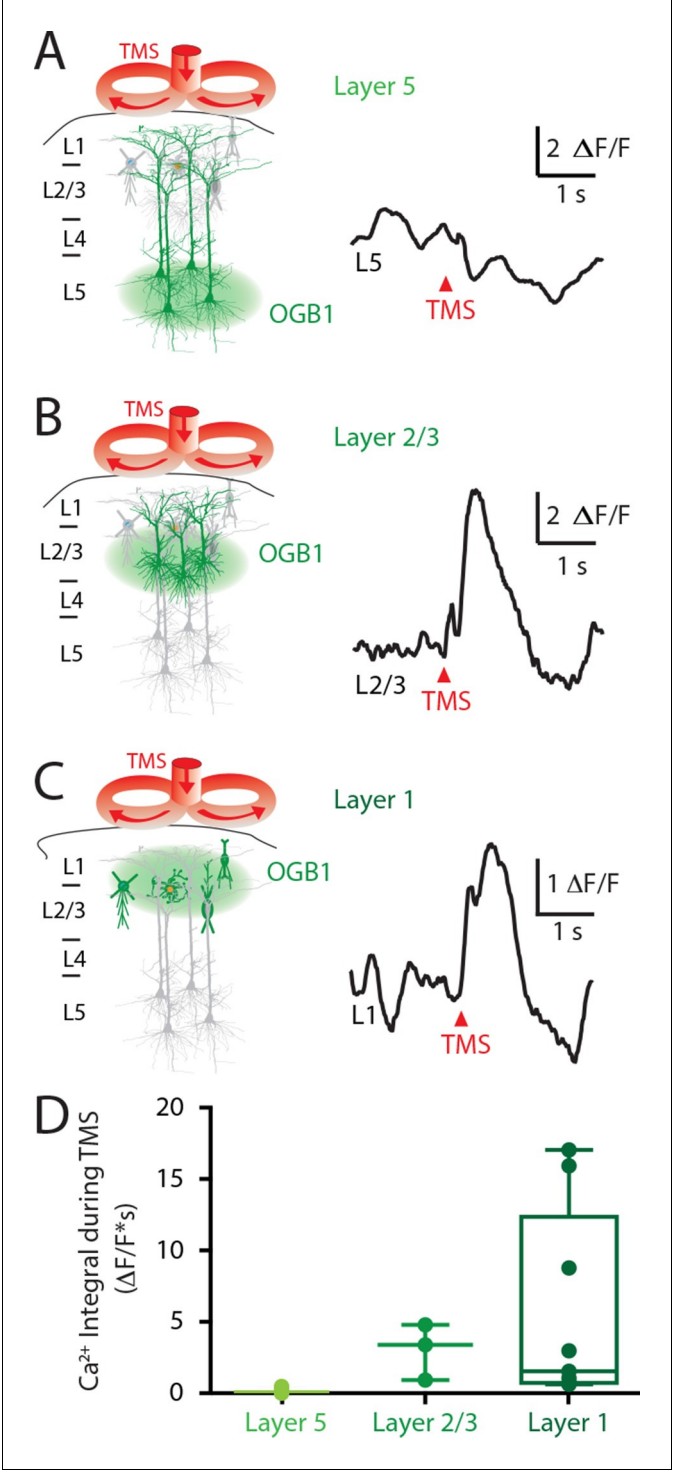

**Figure 3.** Upper layers of the cortex have Ca$^{2+}$ transients in response to TMS. (left) Schematic diagram illustrating Ca$^{2+}$ indicator loaded into (**A**) layer 5, (**B**) layer 2/3 and (**C**) layer 1. For each cortical depth, the Ca$^{2+}$ indicator loading location (green circle) and target neurons (green) are indicated. (right) Ca$^{2+}$ activity was recorded in response to a single TM pulse. (**D**) Comparison of the integrals of the TMS-evoked Ca$^{2+}$ responses recorded at the different cortical depths. TMS, transcranial magnetic stimulation.

To test these possibilities, the $Ca^{2+}$ response to hindpaw stimulation was recorded before (5.5 ± 3.3 $\triangle$F/F•s) and after (3.2 ± 3.6 $\triangle$F/F•s) blocking synaptic transmission by locally applying the AMPA antagonist CNQX to the upper cortical layers at the site of the recording (*Figure 4*). Under these conditions, CNQX prevented the inhibitory effect of TMS in L5 pyramidal neuron dendrites (n = 10; *Figure 4A–C* and *Figure 4—figure supplement 1*). Therefore, since blocking excitatory AMPA-mediated transmission prevented the TMS inhibitory effect, TMS-evoked inhibition in L5 pyramidal neuron dendrites must be of polysynaptic (indirect) origin. We next tested whether the TMS-evoked $Ca^{2+}$ transient in L1 was also of synaptic origin as TMS influenced cell activity in the upper cortical layers (*Figure 4*) and therefore possibly provides the TMS-evoked inhibition of L5 dendrites. Indeed, TMS-evoked activity in L1 neurons was suppressed by CNQX application, significantly reducing the $Ca^{2+}$ response amplitude by 53 ± 7% (n = 8; p<0.05; *Figure 4D–F*). Therefore, the TMS-evoked $Ca^{2+}$ response in L1 neurons is of synaptic origin. Taken together, this data suggests that the inhibition of sensory-evoked L5 dendritic $Ca^{2+}$ responses was primarily mediated by upper-layer inhibitory neurons driven to fire synaptically from neurons or axons recruited by TMS.

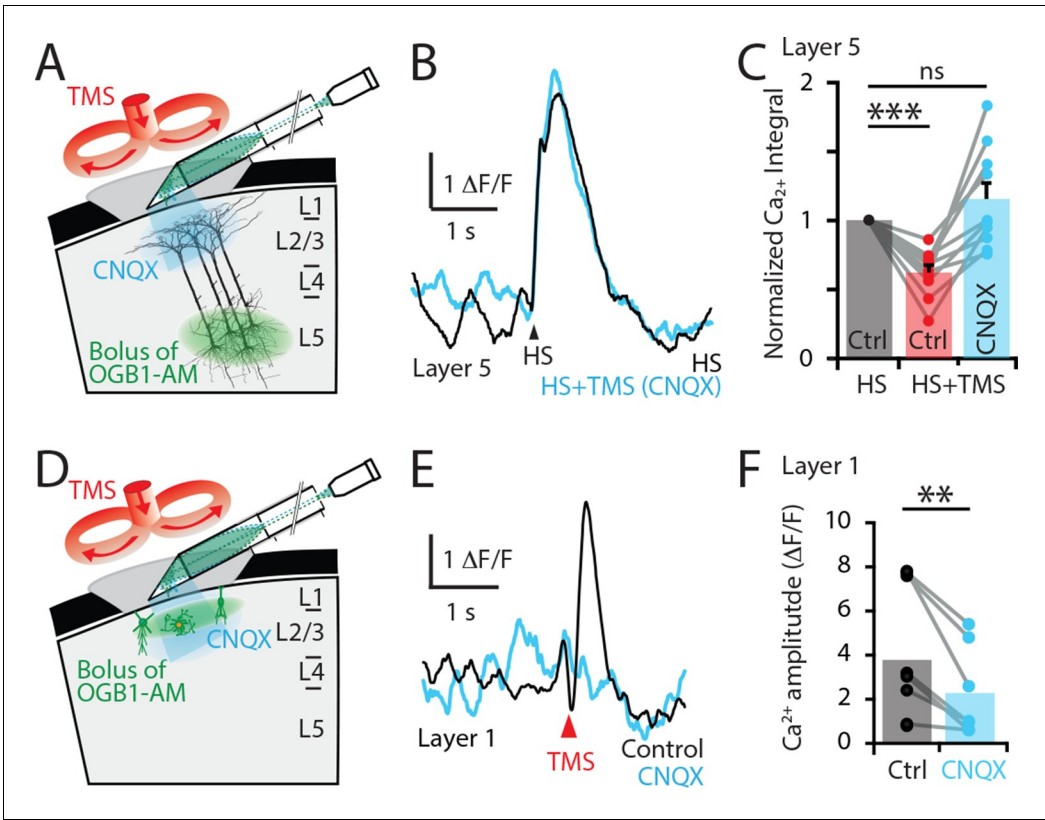

**Figure 4.** TMS directly activates cells in the upper cortical layers. (**A**) Schematic diagram of the experimental design. Layer 5 pyramidal neurons were bulk loaded with OGB1-AM and dendritic $Ca^{2+}$ activity was recorded using a side-on (horizontal) periscope during application of CNQX to the upper cortical layers. (**B**) Typical dendritic $Ca^{2+}$ response to hindpaw stimulation (HS) alone (black) and during a single TMS pulse in the presence of cortical CNQX (blue). (**C**) $Ca^{2+}$ responses (integrals) during HS+TMS in the presence (blue) and absence (red) of CNQX normalized to control HS (black; n=10). (**D**) Schematic diagram of the experimental design. Layer 1 neurons were bulk loaded with OGB1-AM and dendritic $Ca^{2+}$ activity was recorded during TMS using the side-on periscope during application of CNQX into the upper cortical layers. (**E**) Dendritic $Ca^{2+}$ response to a single TMS pulse (black) and in the presence of cortical CNQX (blue). (**F**) Amplitude of the TMS-evoked $Ca^{2+}$ responses in L1 neurons during control (black) and CNQX (blue) (n=8). p<0.005 (**), p<0.001 (***).

The following figure supplement is available for figure 4:

**Figure supplement 1.** Comparison of CNQX application.

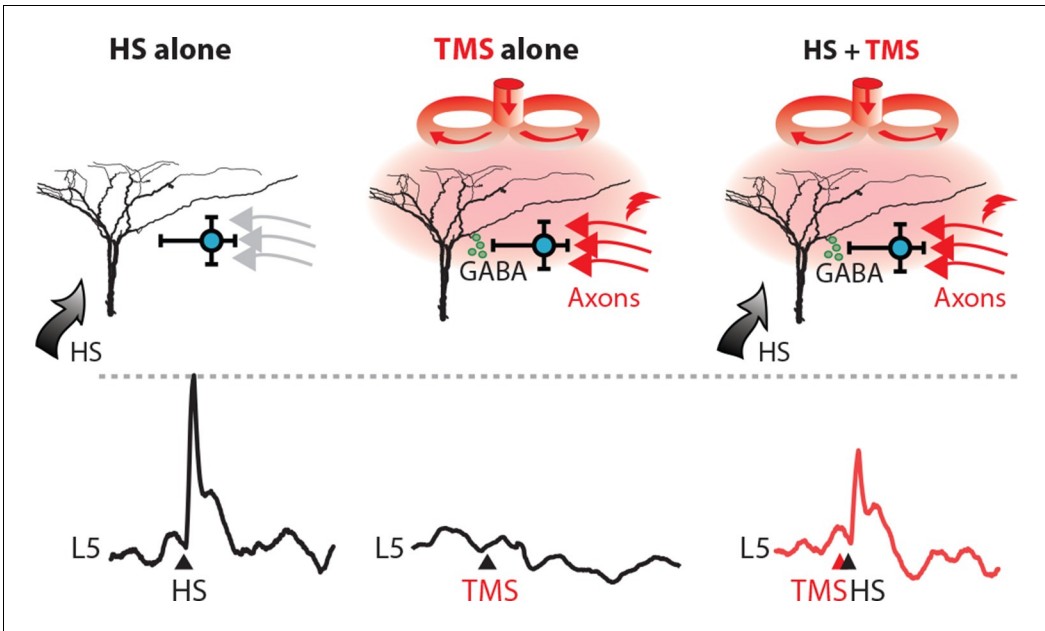

**Figure 5.** TMS activates an inhibitory microcircuit in the upper cortical layers. Hypothesized effect of TMS on cortical processing; TMS activates axons (red) which excite upper layer interneurons (blue) causing GABA neurotransmitter release (green) which provides GABA-mediated inhibition to layer 5 pyramidal neuron dendrites. (left) Hindpaw stimulation (HS) causes large $Ca^{2+}$ responses in layer 5 pyramidal neuron dendrites. (middle) TMS directly activates upper layer neurons but does not cause a $Ca^{2+}$ response in layer 5 dendrites. However, (right) TMS paired with HS causes a large decrease in the HS $Ca^{2+}$ response. TMS, transcranial magnetic stimulation.

## Discussion

The results from this study demonstrate the subcellular effect of TMS on dendritic $Ca^{2+}$ activity for the first time. TMS alone did not directly activate L5 pyramidal dendrites but instead activated axonal processes coursing through the upper layers and synapsing onto $GABA_B$-mediating interneurons in L1. A similar form of synaptic activation of interneurons leading to inhibition in L5 pyramidal neuron dendrites has been shown previously with activation of callosal input to the cortex (*Palmer et al., 2012*). This form of 'silent' inhibition involves the activation of inhibitory conductances which are not detectable at the soma except through their eventual influence on the generation of action potential output. These results therefore highlight an effect of single-pulse TMS on the cortical network, which involves the activation of a specific cortical microcircuit.

The action of TMS at the cellular and network levels is extremely complex and likely constitutes the activation of a range of different cell types leading to multiple effects (*Rossi et al., 2009*; *Siebner et al., 2009*). A recent combined experimental and theoretical investigation of the biophysical underpinnings of TMS suggested that the generation of a magnetic field is most likely to evoke firing in cell bodies as opposed to dendrites or axons (*Pashut et al., 2014*). This is consistent with our finding that no dendritic activity was observed with TMS stimulation alone indicating there was no direct activation of the pyramidal cell dendrites and contrasted with the signals found in both L1 and L2/3 neurons following a TMS pulse. It is also consistent with the activation of neurogliaform interneurons in L1 that target the dendrites of L5 pyramidal neurons and suppress $Ca^{2+}$ activity (*Ziemann, 2010*; *Terao and Ugawa, 2002*; *Pashut et al., 2011*; *Tamás et al., 2003*; *Oláh et al., 2007*; *Oláh et al., 2009*). This form of dendritic inhibition arises from the metabotropic inactivation of L-type ($Ca_V1$) channels in the apical dendrite that underlie the dendritic $Ca^{2+}$ activity (*Wozny and Williams, 2011*) and can last several hundred milliseconds (*Pérez-Garci et al., 2006*). The suppression of $Ca^{2+}$ channels can significantly reduce the action potential firing of the pyramidal neuron even when the driving input to the pyramidal neuron is not predominantly dendritic (*Palmer et al., 2012*). The long time-scale of this form of inhibition (~500 ms) raises interesting consequences for the participation of pyramidal neurons in the cortical network. The similarity of some of the effects of TMS

to interhemispheric inhibition has been noted previously in human studies (*Jiang et al., 2013*; *Lee, 2014*; *Pérez-Garci et al., 2013*; *Ferbert et al., 1992*) including its mediation via GABA$_B$ receptor-activation (*Ferbert et al., 1992*; *Kobayashi et al., 2003*), although these investigations could not examine the cellular mechanisms.

For this study, we used a large coil typically used in humans and a smaller (25 mm) coil designed for use in rodents. The effect on dendritic Ca$^{2+}$ was the same in both cases. Clearly, the use of TMS coils with rats where the magnetic field generated is comparable to the size of the rat brain itself raises the possibility that the effects in humans may differ. However, the cortical feedback fibers which synapse onto the tuft dendrites of pyramidal neurons are located in the part of the cortex closest to the magnetic coil (i.e. L1) in both rats and humans (*Larkum et al., 1999*; *Larkum, 2013*), suggesting there would be overlap with respect to the influence of TMS.

The aim of this study was to examine the effect of TMS on dendritic activity in L5 neocortical neurons rather than a general study on the overall effects of TMS at the cellular level. We were interested in this, in particular, because we have previously hypothesized that cell assemblies over different cortical regions might be associated through the activation of dendritic Ca$^{2+}$ spikes in these neurons (*Larkum, 2013*). According to this hypothesis, dendritic activity in these neurons is a marker of important cognitive processes. The finding that TMS targets this mechanism is therefore highly relevant to the rationale of the study and may be instructive in understanding current applications of TMS. For instance, TMS has been used to alleviate some of the symptoms of hemispatial neglect (*Nyffeler et al., 2009*) and auditory hallucinations (*Giesel et al., 2012*) via unknown inhibitory processes.

In conclusion, the results presented here indicate that the inhibitory actions of TMS is due to the recruitment of upper cortical layer interneurons mediating both GABA$_A$ and GABA$_B$-receptor-activated inhibition in the dendrites of pyramidal neurons. This may have implications for the interpretation of results in humans using TMS as a form of 'virtual lesion' (*Lee et al., 2007*).

## Materials and methods

### Animals and Surgery

Male or female Wistar rats (P30-P40) were used in these experiments. Urethane (intraperitoneal, 1.5 g/kg) was used for experiments under anesthesia, according to the guidelines of the Federal Veterinary Office of Switzerland and LAGeSo Berlin. The head was fixed in a stereotaxic instrument (Model SR-5R, Narishige, Tokyo, Japan) and body temperature maintained at 36 to 37°C. A craniotomy above the primary somatosensory cortex (3 × 4.4 mm square), centered at 1.5 mm posterior to bregma and 2.2 mm from midline in the right hemisphere, was performed. The craniotomy was bathed in normal rat ringer (in mM; 135 NaCl, 5.4 KCl, 1 MgCl$_2$ 1.8 CaCl$_2$, 5 HEPES) and the dura mater surgically removed immediately before Ca$^{2+}$ recording (see below).

### Intrinsic optical imaging

In these vivo experiments, intrinsic optical imaging was used to identify the sensorimotor cortex before surgery. The cortical surface was visualized with green (~530 nm) light to enhance contrast and switched to red (~600 nm) light for functional imaging captured with a charge-coupled device (CCD) camera (Teli) coupled to a 50 mm and 25 mm lens (Navitar). The signal was measured in alternating sweeps before and during contralateral hindpaw stimulation (300 ms; 30 isi) governed by custom routines running in IgorPro (Wavemetrics, Portland, OR.) and Master 8 (A.M.P.I). The intrinsic signal was measured as the difference in the reflected light before and during hindpaw stimulus and was mapped onto the blood vessel pattern to be targeted during experiments.

### Population Ca$^{2+}$ calcium imaging (periscope)

Ca$^{2+}$ imaging was performed as described previously by Murayama et al. (*Murayama et al., 2007*). OGB-1 AM (50 μg; Molecular Probes, Eugene, OR) was mixed with 5 μL of pluronic acid (Pluronic F-127, 20% solution in DMSO, Molecular Probes) for 15 min. The solution was then diluted in 28 μL of HEPES-buffered solution (125 mM NaCl, 2.5 mM KCl, 10 mM HEPES) and mixed for a further 15 min. The OGB-1 AM solution (1.3 mM) was loaded into a glass pipette (tip diameter: 5–15 μm) and pressure-injected into layer 5 (pressure: 10–22 kPa) for 1 min. The pipette was withdrawn and

the area of the craniotomy was then resubmerged with rat ringer for 2 hrs. For epifluorescence $Ca^{2+}$ recordings, light from a blue light-emitting-diode (LED, IBF+LS30W-3W-Slim-RX, Imac Co., Ltd., Shiga, Japan) was passed through an excitation filter, dichroic mirror, and emission filter (as a filter set 31001, Chroma Technology, Rockingham, VT) and focused onto a fiber bundle by a 10× objective (Model E58-372, 0.45 NA, Edmund Optics GmbH, Germany). The fiber bundle (IGN-06/17, Sumitomo Electric Industries, Tokyo, Japan) was used as a combined illuminating/recording fiber and consisted of 17,000 fiber elements. The end face of the bundle was fitted with a prism-lens assembly, which consisted of a right-angle prism (dimension of $0.5 \times 0.5 \times 0.5$ mm, GrinTech, Jena, Germany) attached to a GRIN lens (a diameter of 0.5 mm and a NA of 0.5, GrinTech). In previous studies, the fiber optic 'periscope' was vertically inserted 500 µm into the brain at a 90° angle (*Murayama et al., 2009*). This ensured that the $Ca^{2+}$ responses were recorded from the upper cortical layers. However, in this study, the fiber optic could not be inserted vertically due to the positioning of the TMS coil. Instead, the fiber optic was positioned horizontally on the brain surface ('flat' periscope). In this configuration, the 'flat' periscope was able to capture the same $Ca^{2+}$ responses as the 'vertical' periscope. As previously reported by (*Murayama et al., 2009*), TTX application into L5 caused a dramatic increase in L5 dendritic $Ca^{2+}$ responses to hindpaw stimulation using both the 'flat' periscope and the 'vertical' periscope (*Figure 1—figure supplement 1*). With a focal length nominally 100 µm and 0.73 × magnification (*Murayama et al., 2007*), the flat periscope configuration resulted in a field of view of 685 µm diameter restricted to the upper layers of the cortex. A cooled CCD camera operating at either 100 Hz (MicroMax, Roper Scientific, Trenton, NJ) or 2.7 kHz (Redshirt imaging, Decatur, GA) was used for collecting fluorescence. The fluorescence signals were quantified by measuring the mean pixel value of a manually selected ROI for each frame of the image stack using Igor software. Data was acquired on a PC using WinView software (Roper Scientific). Regions of interest (ROIs) were chosen offline for measuring fluorescence changes (see 'Data analysis').

## Transcranial magnetic stimulation

TMS was applied to the rat somatosensory cortex using a MagStim 200 Monopulse and Rapid 2 system (The MagStim Company Ltd., Whitland, UK) figure-eight coil, which was positioned 2–3 cm from the brain using a fixed manipulator. Experiments were typically performed with a 70-mm coil (exception: during CNQX application, TMS was delivered via a 25-mm coil, see *Figure 1—figure supplement 2*). Figure-of-eight-shaped coil was used as they produce a more focal current which is maximal at the intersection of the two round components (*DeFelipe, 2011*). The coil was centered on the craniotomy directly above the periscope fiber optic cable and angled approximately parallel to the skull curvature. TTL digital pulses triggered a single pulse TMS at 80–100% stimulation intensity (unless otherwise stated) with an inter-trial interval of at least 9 s to limit fluorescence bleaching. Given this experimental design, the electric field should be approximately ~150–200 V/m (*Cohen et al., 1990*) magnetic stimulation is comparatively indifferent to the conductive properties of the skull (*Wagner et al., 2006*), and since the small ($3 \times 4.4$ mm) craniotomy is therefore unlikely to change the currents produced by the coil. Further, there was no behavioral response of the rat during TMS and increasing TMS strength did not elicit an excitatory response in layer 5 (*Figure 1—figure supplement 3*). When paired with hindpaw stimulus, the TMS was activated 50 ms before the contralateral hindpaw stimulus (> 10 trials per animal). Electrical stimulation of the contralateral hindpaw was achieved by applying a brief (1 ms; 100 V) current onto conductive adhesive strips (approximately 1 cm wide by 2 cm long) placed on the contralateral hindpaw pad. Where stated, hindpaw stimulation was also achieved by a triggered airpuff delivered to within 1 cm of the hindpaw (~40 psi; ~400 ms). All trials were interleaved in each experiment to limit time-dependent effects.

## Drug application

The $GABA_B$ receptor antagonist CGP52432 (1 µM; Tocris) and the $GABA_A$ receptor antagonist Gabazine (3 µM; Tocris) were applied to the cranial surface, AMPA/kainate receptor antagonist CNQX (50 µM) was either applied to the cranial surface or pressure injected into layer 2/3. TTX (1 µM) was pressure injected into layer 5. See *Figure 2—figure supplement 1* for cortical spread of drug application.

## Two-photon imaging to assess cortical penetration of drug application

The penetration of pressure injected or cortically applied drugs was measured using in vivo two-photon microscopy (see *Figure 2—figure supplement 1*). Brain tissue was imaged to a depth of 500 μm using a two-photon microscope (Thorlabs A-scope) with a titanium sapphire laser (860 nm; 140 fs pulse width; SpectraPhysics MaiTai Deepsee) passed through a 40x water immersion objective (Olympus; 0.8 NA). Images were obtained in full-frame mode (512 x 512 pixels).

## Data analysis

The fluorescence signals in vivo were quantified by measuring the mean pixel value of a manually selected (offline) ROI for each frame of the image stack using IgorPro software (Wavemetrics). ROIs included the entire field of view and $Ca^{2+}$ changes were expressed as $\Delta F/F = F_t / F_0$, where $F_t$ was the average fluorescence intensity within the ROI at time t during the imaging experiment and $F_0$ was the mean value of fluorescence intensity before stimulation. $Ca^{2+}$ responses were measured as the maximum value (amplitude) and total area under the positive trace (integral) within 1 s of the hindpaw stimulation. All numbers are indicated as mean ± s.e.m. Significance was determined using parametric tests (paired/unpaired student t-test) or non-parametric tests (Unpaired, Mann-Whitney test; paired, Wilcoxon matched-pairs signed rank test) as appropriate. $p < 0.05$ (*), $p < 0.01$ (**) and $p < 0.001$ (***).

## Acknowledgements

This work was supported by SystemsX.ch (NeuroChoice). We would like to acknowledge the software assistance of Juan Nunez-Iglesias at the Victorian Life Sciences Computation Initiative. We would also like to thank Rogier Min, Julie Seibt and Adam Shai for technical assistance and discussions.

## Additional information

### Funding

| Funder | Grant reference number | Author |
| --- | --- | --- |
| SystemsX.ch (NeuroChoice) | | Sean C Murphy<br>Lucy M Palmer<br>Matthew E Larkum |
| National Health and Medical Research Council | 1085708 | Lucy M Palmer |
| National Health and Medical Research Council | 1063533 | Lucy M Palmer |
| Deutsche Forschungsgemeinschaft | EXC 257 NeuroCure | Matthew E Larkum |
| Schweizerischer Nationalfonds zur Förderung der Wissenschaftlichen Forschung | 31003A_130694 | Matthew E Larkum |

The funders had no role in study design, data collection and interpretation, or the decision to submit the work for publication.

### Author contributions

SCM, Conception and design, Acquisition of data, Analysis and interpretation of data, Drafting or revising the article; LMP, MEL, Conception and design, Analysis and interpretation of data, Drafting or revising the article; TN, RMM, Conception and design, Drafting or revising the article, Contributed unpublished essential data or reagents

### Author ORCIDs

Matthew E Larkum, http://orcid.org/0000-0001-9799-2656

## Ethics

Animal experimentation: All experiments were conducted in strict accordance with the guidelines of the Federal Veterinary Office of Switzerland and LAGeSo Berlin (Ethics number: GO257/11).

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
