## [Decision Letter]

Thank you for submitting your work entitled "Transcranial Magnetic Stimulation (TMS) inhibits cortical dendrites" for consideration by *eLife*. Your article has been favourably assessed by Timothy Behrens (Senior Editor) and two reviewers, one of whom, Marlene Bartos, is a member of our Board of Reviewing Editors, and another is Alon Korngreen.

The reviewers have discussed the reviews with one another and the Reviewing Editor has drafted this decision to help you prepare a revised submission.

Summary:

Both reviewers evaluated your study on the cellular mechanisms underlying the TMS-mediated effects on the activity of layer 5 Pyramidal cells of the somatosensory cortex as highly relevant for the neuroscience community. TMS stimulation is used as non-invasive technology in humans for the treatment of various diseases including epilepsy and schizophrenia. It is therefore pivotal to understand how TMS influences cortical neuronal network activity. Despite the high relevance of the study some major criticisms have been formulated with the aim to improve the value of the work.

Essential revisions:

Major criticism: First, the usage of TMS requires a precise calibration of the stimulus intensity above the motor cortex. Intensity is increased until the contralateral limb is moved. It is therefore important to add controls or calibrations to the study ('Materials and methods' section). Refinement of TMS strength will have a high impact on the activated neuron types. Interneurons have a lower a spike threshold during TMS stimulation than principal cells. Thus, setting the stimulus intensity will have a large impact on the results. Second, the size of the coils used in the study is usually applied for human studies. This can be considered as focal stimulation for human cortex. However, it may be far from focal for a rat cortex. Thus, the authors should address the issues of locality by changing the location of magnetic stimulation while recording from the same cortical location. Finally, the conclusions from the experimental results point to the direct activation of GABAergic cells by TMS. The manuscript would be markedly improved if the direct activation of GABAergic interneurons could be shown. Finally, the study was focused on GABA_B_ receptor-mediated actions. However, a contribution of GABA_A_ receptors should also be expected and thus discussed in the manuscript. A discussion on possible pre- versus postsynaptic effects of GABA_B_ in relation to the obtained results should be discussed.

*Reviewer #1:*

Murphy et al. examine in their study the cellular mechanisms underlying the TMS stimulation-mediated effects in layer 5 Pyramidal cells of the somatosensory cortex. By using fiber optics and Ca^2+^ imaging in vivo, the authors showed that TMS stimulation reduced Ca^2+^ activity in distal dendrites of Pyramidal cells evoked by tactile stimuli. This suppression could be abolished by GABA_B_ receptor blockers, or by abolishing dendritic synaptic excitatory transmission. TNS is an important non-invasive tool in humans and used for the treatment of various diseases including epilepsy and schizophrenia. Therefore it is important to understand what precisely TMS is doing to the activity of feedback projections.

Major criticisms:

1) It remained unclear why the planned examination of Ca^2+^ transients in Pyramidal cell dendrites during application of the GABA_A_ receptor antagonist Gabazine in combination with TMS stimulation was impossible. If the authors believe that blocking GABARs induced such a strong activation of Ca^2+^ channels that TMT could not result in detectable changes in the Ca^2+^ transients than lower concentrations of Gabazine could have been tested. Please include an explanation on why this experiment was not possible.

2) Although the evidences/conclusion from the experiments are plausible and point to the direct activation of GABAergic cells by TMS, it would be great if the authors could directly show that this is indeed is the case by using for example a transgenic mouse line in which GABAergic cells are tagged with tdT (GAD65/67-Cre & viral injection of a construct which allows tdT expression under Cre control) to visually identify interneurons in the upper layers and to obtain 2P imaging from the interneuron activity during tactile stimulation with and without TMS application. Please consider whether this proposed experiment or another one directly testing interneuron recruitment wit TMS would be possible in a short time window.

3) The Discussion should include a chapter on the possible cellular mechanisms by which TMS selectively recruits GABAergic cells.

4) Is the effect of the GABA_B_ receptor blocker pre- or postsynaptic? Please discuss possible contributions of both cellular mechanisms on the observed effects.

*Reviewer #2:*

Murphy et al. combine for the first time transcranial magnetic stimulation (TMS) and in-vivo calcium imaging to record neuronal activity from superficial cortical layers during magnetic stimulation. The authors clearly show that TMS triggers neuronal activity in cortical neurons. They present evidence arguing that the activity recorded from dendrites in L1 is due to poly synaptic activation of superficial inhibitory interneurons. This is an important step towards a mechanistic understanding of magnetic stimulation of the nervous system. Unfortunately, due to the way the authors use, or rather abuse, TMS the manuscript is out of context with the general body of scientific knowledge. This manuscript can potentially be of importance to the TMS community. I think the authors may have a true impact on the field if they perform some additional experiments and change the focus of the manuscript.

The paramount problem, as I see it, is the experimental design. The authors use the TMS device without regard to its proper use. In a standard TMS experiment stimulation intensity is first calibrated over the motor cortex. The intensity is increased until the contralateral limb moves. This is called the motor threshold and all results in the TMS field are normalized to this value which changes from subject to subject. The motor threshold is not an abstract point. It has been shown that responses below motor threshold are often inhibitory whereas exceeding motor threshold is considered excitatory. Several in-vitro studies (Rotem and Moses Biophys J 2008, Bonmassar, Nat.Commun. 2012, Pashut, Front. Cell. Neuro. 2014) have shown that low intensity stimulation stimulates first neurons with low action potential threshold. In the cortex, these are probably low threshold interneurons.

In contrast to the available knowledge and established procedures the authors used the Magstim at 100%. No explanation is given for these settings. This setting is meaningless when it is not calibrated to motor threshold. Since the authors did not detect any behavioural changes that were induced by TMS I suspect that they were below motor threshold. This directly leads to their observation of the inhibitory effect of TMS. To allow comparing their study to the general TMS literature it is imperative that they calibrate their TMS device by observing limb movement or by recording MEP from the limb's muscle. Many human TMS studies predict that as the intensity increases there will be larger activation of neurons in deeper layers of the cortex. At motor threshold it is possible to predict that L5 pyramidal neurons will fire. The authors are uniquely poised to demonstrate these effects with only a few more experiments and proper calibration of the stimulus.

Another problematic point is the use of the 70 mm and 25 mm butterfly coils. These coils were designed for humans. The largest magnetic field (at the intersection of the two coils) occupies, roughly, a volume of 10 mm cubed. This can be considered as focal stimulation for human cortex. However, it is far from focal for a rat cortex. Thus, the authors should address the issues of locality by changing the location of magnetic stimulation while recording from the same cortical location. I predict that they will observe effects quite similar to those they already report because their stimulus is so distributed.

The logic presented by the authors is somewhat circumvent. TMS is just another way to electrically stimulate neurons in the brain. Thus, it triggers action potentials leading to release of neurotransmitter. Why focus on GABA_B_? It is clear that GABA is released activating both GABA_A_ and GABA_B_ receptors. The authors point this later in the manuscript. However, placing the metabotropic aspect of the stimulus in the front weakens their argument. This will further be weakened if indeed, at higher TMS intensities, they will observe excitation. As it is written the reader may think that your primary interest in cortical inhibition and not explaining how TMS excites cortical tissue (which is what you end up showing).

---

## [Author Response]

Essential revisions:

*Major criticism: First, the usage of TMS requires a precise calibration of the stimulus intensity above the motor cortex. Intensity is increased until the contralateral limb is moved. It is therefore important to add controls or calibrations to the study (Materials and methods section). Refinement of TMS strength will have a high impact on the activated neuron types. Interneurons have a lower a spike threshold during TMS stimulation than principal cells. Thus, setting the stimulus intensity will have a large impact on the results.*

We have added controls to the manuscript including a new figure supplement which illustrates the Ca^2+^ response to increasing TMS strength in layer 5 pyramidal neurons (Figure 1—figure supplement 3).

*Second, the size of the coils used in the study is usually applied for human studies. This can be considered as focal stimulation for human cortex. However, it may be far from focal for a rat cortex.*

Although the larger 70 mm coil is indeed typically applied for human studies, the smaller 25 mm coil we use is designed for use with rodents and is even described by the manufacturer, Magstim, as the “rat coil” (despite the claim of the 2nd reviewer). The influence of the different coils were investigated and were not significantly different (Figure 1—figure supplement 2). The spatial extent of the stimulation is discussed in the text and is not referred to as focal.

*Thus, the authors should address the issues of locality by changing the location of magnetic stimulation while recording from the same cortical location.*

We now include a figure (Figure 1—figure supplement 3), which illustrates the influence of the TMS coil location on the sensory-evoked dendritic response. TMS-evoked inhibition of dendritic sensory responses was only measurable when the TMS coil was located near (within 10mm) to the somatosensory cortex.

*Finally, the conclusions from the experimental results point to the direct activation of GABAergic cells by TMS. The manuscript would be markedly improved if the direct activation of GABAergic interneurons could be shown.*

The experimental results illustrate that GABAergic cells are active during TMS. This is illustrated firstly in Figure 3 which clearly shows a large Ca^2+^ response in layer 1 (which consists of only interneurons) during TMS. We further show that this is not due to direct activation of these interneurons by TMS, but is because TMS activates axons which synapse onto these interneurons (Figure 4).

*Finally, the study was focused on GABA_B_ receptor-mediated actions. However, a contribution of GABA_A_ receptors should also be expected and thus discussed in the manuscript.*

We now include a new figure (Figure 2—figure supplement 2), which illustrates that the application of the GABA_A_ receptor blocker, Gabazine, during TMS does not cause additional dendritic activation over and above GABA_B_ block with CGP52432. As evident from the large increase in the sensory evoked dendritic response in the presence of a GABA_A_ blocker (see Figure 2—figure supplement 2), GABA_A_ inhibition is tonically active. This is now further discussed in the manuscript.

*A discussion on possible pre- versus postsynaptic effects of GABA_B_ in relation to the obtained results should be discussed.*

The possible pre- and postsynaptic effects of TMS-evoked GABA_B_ mediated dendritic inhibition is now discussed in the manuscript.

Reviewer #1:

*Major criticisms:*

*1) It remained unclear why the planned examination of Ca^2+^ transients in Pyramidal cell dendrites during application of the GABA_A_receptor antagonist Gabazine in combination with TMS stimulation was impossible. If the authors believe that blocking GABARs induced such a strong activation of Ca^2+^ channels that TMT could not result in detectable changes in the Ca^2+^ transients than lower concentrations of Gabazine could have been tested. Please include an explanation on why this experiment was not possible.*

We actually did do this experiment but decided the data was relatively uninformative so didn’t include it. In hindsight, we agree with the reviewer that the reader should be able to see the results. GABA_A_ block significantly increased the dendritic Ca^2+^ response to the control hindpaw stimulation, as we showed in a previous publication (Murayama et al., 2009). Since this pharmacological manipulation severely alters dendritic Ca^2+^ dynamics regardless of TMS stimulation, its use for investigating the effect of TMS on GABA_A_ activation is problematic. That said, we now include an additional supplementary figure where we have applied Gabazine to the cortical surface and we find there is no additional effect above the GABA_B_ block (see Figure 2—figure supplement 2). Furthermore, block of GABA_B_ receptors, while not significantly affecting the Ca^2+^ response to hindpaw stimulation completely occluded the effect of TMS which we take to indicate that the TMS effect on dendritic Ca^2+^ is either predominantly or entirely through GABA_B_ receptors.

*2) Although the evidences / conclusion from the experiments are plausible and point to the direct activation of GABAergic cells by TMS, it would be great if the authors could directly show that this is indeed is the case by using for example a transgenic mouse line in which GABAergic cells are tagged with tdT (GAD65/67-Cre & viral injection of a construct which allows tdT expression under Cre control) to visually identify interneurons in the upper layers and to obtain 2P imaging from the interneuron activity during tactile stimulation with and without TMS application. Please consider whether this proposed experiment or another one directly testing interneuron recruitment wit TMS would be possible in a short time window.*

Firstly, the experiments illustrate that GABAergic cells are not directly activated by TMS in the region we are recording, and their activity during TMS is due to synaptic input from directly activated axons (see Figure 4 which shows a significant decrease in the TMS-evoked calcium response in layer 1 neurons – which are all GABAergic interneurons – in the presence of CNQX).

Secondly, the proposed experiments would be wonderful if they were possible – however, technically, it is not possible to put a TMS coil over the craniotomy during 2P imaging as the 2P objective (and associated laser path) would be in the way.

*3) The Discussion should include a chapter on the possible cellular mechanisms by which TMS selectively recruits GABAergic cells.*

We now make it clearer in the Discussion that GABAergic cells are recruited via the direct activation of axons coursing through the upper layers of the cortex.

*4) Is the effect of the GABA_B_ receptor blocker pre- or postsynaptic? Please discuss possible contributions of both cellular mechanisms on the observed effects.*

Our previous findings have illustrated that the GABA_B_ -mediated dendritic inhibition evoked by upper layer fibers is due to the activation of postsynaptic GABA_B_ receptors (Perez-Garci et al., 2006; Palmer et al., 2012). We have no reason to expect different mechanisms are at play in this study, since a similar micro-network is activated. That said, we now include the possible contributions of pre- and postsynaptic GABA_B_ receptors during TMS in the manuscript.

Reviewer #2:

*[…] The paramount problem, as I see it, is the experimental design. The authors use the TMS device without regard to its proper use. In a standard TMS experiment stimulation intensity is first calibrated over the motor cortex. The intensity is increased until the contralateral limb moves. This is called the motor threshold and all results in the TMS field are normalized to this value which changes from subject to subject. The motor threshold is not an abstract point. It has been shown that responses below motor threshold are often inhibitory whereas exceeding motor threshold is considered excitatory. Several in-vitro studies (Rotem and Moses Biophys J 2008, Bonmassar, Nat.Commun. 2012, Pashut, Front. Cell. Neuro. 2014) have shown that low intensity stimulation stimulates first neurons with low action potential threshold. In the cortex, these are probably low threshold interneurons.*

*In contrast to the available knowledge and established procedures the authors used the Magstim at 100%. No explanation is given for these settings. This setting is meaningless when it is not calibrated to motor threshold. Since the authors did not detect any behavioural changes that were induced by TMS I suspect that they were below motor threshold. This directly leads to their observation of the inhibitory effect of TMS.*

We now include a new figure (Figure 1—figure supplement 3), which illustrates the Ca^2+^ response in Layer 5 pyramidal neuron dendrites to different intensities of TMS. An excitatory response was not observed in layer 5 pyramidal neuron dendrites at any intensity. Our results show inhibition of sensory-evoked responses by TMS in layer 5 pyramidal neuron dendrites, however, as illustrated in Figure 3, layer 2/3 neurons have an excitatory response. Due to the necessary imprecision of the stimulation (given the comparative sizes of the coil and the rat brain), the motor cortex would be expected to be affected similarly to the somatosensory cortex in our experiments particularly since these cortical areas overlap in rats. The reviewer is correct that TMS pulses were below motor threshold as we never detected movement. Nevertheless, we also saw no evidence for direct activation of low threshold inhibitory neurons as evidenced by the complete occlusion using CNQX.

*To allow comparing their study to the general TMS literature it is imperative that they calibrate their TMS device by observing limb movement or by recording MEP from the limb's muscle. Many human TMS studies predict that as the intensity increases there will be larger activation of neurons in deeper layers of the cortex. At motor threshold it is possible to predict that L5 pyramidal neurons will fire. The authors are uniquely poised to demonstrate these effects with only a few more experiments and proper calibration of the stimulus.*

We do not induce motion or limb movement while using maximum TMS intensity and therefore we are unable to increase the TMS strength to induce excitation in layer 5 pyramidal neurons. This may be because our experiments are conducted under urethane anaesthesia. This is necessary to be able to record Ca^2+^ transients during hindpaw stimulation and TMS in vivo, however, this may occlude any expected limb movements.

*Another problematic point is the use of the 70 mm and 25 mm butterfly coils. These coils were designed for humans. The largest magnetic field (at the intersection of the two coils) occupies, roughly, a volume of 10 mm cubed. This can be considered as focal stimulation for human cortex. However, it is far from focal for a rat cortex. Thus, the authors should address the issues of locality by changing the location of magnetic stimulation while recording from the same cortical location. I predict that they will observe effects quite similar to those they already report because their stimulus is so distributed.*

The 25 mm butterfly coil was designed for use in animal models and is even described by the manufacturer (Magstim) as the “rat coil” – that said, we did not see influence of coil size on the effect of TMS on the sensory evoked Ca^2+^ response (see Figure 1—figure supplement 2). Further, we now include a figure illustrating the effect of moving the TMS coil to different locations (Figure 1—figure supplement 3).

*The logic presented by the authors is somewhat circumvent. TMS is just another way to electrically stimulate neurons in the brain. Thus, it triggers action potentials leading to release of neurotransmitter. Why focus on GABA_B_? It is clear that GABA is released activating both GABA_A_ and GABA_B_ receptors. The authors point this later in the manuscript. However, placing the metabotropic aspect of the stimulus in the front weakens their argument. This will further be weakened if indeed, at higher TMS intensities, they will observe excitation. As it is written the reader may think that your primary interest in cortical inhibition and not explaining how TMS excites cortical tissue (which is what you end up showing).*

Our primary interest with these experiments was to examine the effect of TMS on dendritic activity. While this may not be the reviewer’s main interest, we were nevertheless excited to find that TMS lead to a down-regulation of Ca^2+^ activity in L5 pyramidal dendrites without any direct stimulation of these neurons. This points to a specificity of TMS that was unexpected and is potentially extremely useful. The reviewer asks, “Why the focus on GABA_B_? It is clear that GABA is released activating both GABA_A_ and GABA_B_ receptors”. As we have shown here and in other studies (cf Palmer et al., 2012), counter-intuitively the primary effect of stimulating certain projections to

L1 can be the activation of specific dendrite-targeting, GABA_B_ mediating inhibitory neurons. This cell type (the neurogliaform neuron) has been well characterized and predominantly mediates its effect through GABA_B_ receptors probably due to volume transmission. Furthermore, activation of GABA_B_ receptors has a remarkably specific effect blocking dendritic L-type Ca^2+^ channels (cf Perez-Garci et al., 2013). Here we report that TMS appears to recruit this sub-population of cells and that blocking GABA- B receptors completely occluded the effect of TMS on dendritc Ca^2+^ activity. Given this, it would seem unwise to assume that GABA_A_ receptors played any significant role in suppressing dendritic Ca^2+^.

On the other hand, the experiments do suggest that TMS could be used as a tool to investigate dendritic excitability non-invasively in other preparations (e.g. humans) that may give us a window into this otherwise tricky subject. Of course, further tests need to be made for the extension of these results to other preparations, and these were clearly out of the scope of this study, but the implications of these initial data in rodents could, in our opinion, be enormous. If the reviewer is correct, that under other conditions the situation might change (“at higher TMS intensities, they will observe excitation”), this would not change the prediction that TMS can in principle be used to down-regulate dendritic Ca^2+^ activity non-invasively (e.g. at lower intensities).